# Evolving Skeletons: Motion Dynamics in Action Recognition

Jushang Qiu
Australian National University
Canberra, Australian Capital Territory, Australia
jushang.qiu@anu.edu.au

Lei Wang*
Griffith University
Brisbane, Queensland, Australia
Australian National University
Canberra, Australian Capital Territory, Australia
l.wang4@griffith.edu.au

## Abstract

Skeleton-based action recognition has gained significant attention for its ability to efficiently represent spatiotemporal information in a lightweight format. Most existing approaches use graph-based models to process skeleton sequences, where each pose is represented as a skeletal graph structured around human physical connectivity. Among these, the Spatiotemporal Graph Convolutional Network (ST-GCN) has become a widely used framework. Alternatively, hypergraph-based models, such as the Hyperformer, capture higher-order correlations, offering a more expressive representation of complex joint interactions. A recent advancement, termed Taylor Videos, introduces motion-enhanced skeleton sequences by embedding motion concepts, providing a fresh perspective on interpreting human actions in skeleton-based action recognition. In this paper, we conduct a comprehensive evaluation of both traditional skeleton sequences and Taylor-transformed skeletons using ST-GCN and Hyperformer models on the NTU-60 and NTU-120 datasets. We compare skeletal graph and hypergraph representations, analyzing static poses against motion-injected poses. Our findings highlight the strengths and limitations of Taylor-transformed skeletons, demonstrating their potential to enhance motion dynamics while exposing current challenges in fully using their benefits. This study underscores the need for innovative skeletal modeling techniques to effectively handle motion-rich data and advance the field of action recognition.

## CCS Concepts

• **Computing methodologies** → *Machine learning algorithms*; **Activity recognition and understanding**; • **Networks** → *Network design principles*.

## Keywords

Spatiotemporal, Graph, Hypergraph, Motion, Evaluation

### ACM Reference Format:

Jushang Qiu and Lei Wang. 2025. Evolving Skeletons: Motion Dynamics in Action Recognition. In *Companion Proceedings of the ACM Web Conference 2025 (WWW Companion '25), April 28-May 2, 2025, Sydney, NSW, Australia.* ACM, New York, NY, USA, 22 pages. https://doi.org/10.1145/3701716.3717739

*Corresponding author.

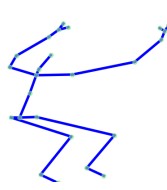 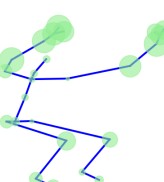

Figure 1: Visual comparison of (*Left*) a static pose and (*Right*) a Taylor-transformed skeleton for the action *cheer up*. Motion dynamics are overlaid on the original skeleton for enhanced visualization. The size of the green circles indicates motion intensity, with larger circles representing bigger motions. Taylor-transformed skeletons emphasize dominant motions and dynamic patterns, while static poses highlight spatial arrangements of the joints. Additional visualizations are provided in the Appendix.

## 1 Introduction

Skeletal action recognition has seen significant progress due to its ability to represent human motion in a structured manner, with applications ranging from surveillance to healthcare and entertainment [10, 22, 47, 65, 68, 82–84, 87, 88, 92, 93, 111]. However, as the field matures, new challenges continue to emerge. While traditional skeleton-based methods [5, 13, 44, 82, 84, 98, 104, 106, 114, 116] excel at encoding the static spatial relationships of body joints, they often struggle to capture the full spectrum of dynamic motion[19, 20, 128], particularly in complex and highly coordinated actions.

Addressing this challenge requires moving beyond simple static representations and incorporating richer, motion-centric features that can better model the temporal evolution of human actions [5, 7–9, 13, 44, 62, 67, 104]. The recent rise of graph-based models, particularly Spatial-Temporal Graph Convolutional Networks (ST-GCN)[110], has provided a powerful framework for skeleton-based action recognition by using the relationships between body joints over both space and time. These models represent human skeletons as graphs, where joints are nodes and their interactions, whether spatial or temporal, are encoded as edges. While effective in modeling pairwise dependencies between joints, such graph-based methods [42, 50, 60, 64, 84, 102, 121] still face limitations when attempting to capture complex, higher-order relationships, particularly in actions where the coordination between multiple joints is essential.

In response to these limitations, more advanced techniques have been introduced, such as Taylor-transformed skeleton sequences[95], which extend traditional representations by embedding motion

dynamics through higher-order temporal derivatives (Figure 1). This transformation introduces additional layers of information, such as velocity and acceleration, which enhance the temporal representation of human motion and are particularly useful for distinguishing between similar actions that differ in their dynamics. Furthermore, hypergraph-based models, such as Hyperformer[120], have emerged as a promising approach to address the complexity of joint interactions by representing not only pairwise relationships but also higher-order joint dependencies through hyperedges[2, 11, 33, 63, 81, 89, 126].

Despite the promising potential of these approaches, there has been limited work comparing their effectiveness across different domains and contexts. This paper aims to fill this gap by evaluating three key aspects of skeleton-based action recognition: (1) the comparison between ST-GCN and Hyperformer in their ability to model both spatial and temporal relationships, (2) the examination of traditional skeleton sequences versus Taylor-transformed skeleton sequences in terms of their ability to capture dynamic motion, and (3) the evaluation of skeletal graphs versus hypergraphs in their capacity to represent joint dependencies and action dynamics. The **contributions** of this paper are as follows:

i. We provide a comparative evaluation of ST-GCN and Hyperformer, shedding light on their respective strengths and weaknesses in modeling complex human actions, with a focus on their ability to capture spatial, temporal, and higher-order joint dependencies.

ii. We investigate the impact of using traditional skeleton sequences versus Taylor-transformed skeleton sequences, highlighting how the inclusion of motion dynamics (such as velocity) improves action recognition, particularly for dynamic and intricate movements.

iii. We analyze the trade-offs between skeletal graphs and hypergraphs in representing the relationships between joints, offering insights into the conditions under which each approach excels, and providing practical guidance for selecting the most suitable model for specific action recognition tasks.

## 2 Related Work

Below, we review the most closely related works, categorizing them by key methodologies, and highlighting the novel contributions of our work in evaluating these different approaches.

**Graph-based models.** The use of graph-based models, particularly ST-GCN, has become a dominant approach for skeleton-based action recognition. In ST-GCN [110], the human skeleton is represented as a graph where joints serve as nodes and the connections between them as edges. The edges model spatial dependencies, while temporal dependencies are captured by treating the skeleton sequence as a dynamic graph over time. ST-GCN has been shown to be highly effective for action recognition, achieving state-of-the-art performance on several benchmark datasets such as NTU RGB+D [56, 71] and Kinetics [6]. Several improvements on the original ST-GCN have been proposed to address its limitations[24]. These include methods that incorporate attention mechanisms [15, 32, 48, 66, 80, 105, 117], multi-scale graph convolutions [4, 23, 30, 36, 49, 78, 101, 107, 113, 127], and feature fusion strategies [17, 40, 42, 61, 62, 75]. However, these models typically

rely on static representations of motion, where temporal evolution is embedded indirectly within the graph structure. As a result, they often struggle to capture the more complex, dynamic motion transitions between joints, which are crucial for distinguishing between certain types of actions, such as running *vs.*walking.

While ST-GCN remains one of the foundational models in this domain, our work differentiates itself by directly evaluating its performance in comparison with other representations, specifically Taylor-transformed skeletons. We assess how enriching skeleton sequences with higher-order temporal derivatives improves action recognition, especially for actions involving complex motion dynamics. By contrasting ST-GCN with Hyperformer [120], we provide new insights into how hypergraph-based models [3, 16, 39, 43, 51, 103, 125], which consider higher-order joint dependencies, can outperform traditional skeletal graphs in certain tasks.

**Motion-centric approaches.** The limitation of static representations of skeleton sequences in ST-GCN models has led to the development of motion-centric representations [14, 45, 53, 95, 97, 122], such as Taylor-transformed skeleton sequences [95]. These transformations use higher-order temporal derivatives (*e.g.*, velocity and acceleration) to enrich skeleton representations by emphasizing dynamic motion patterns. As introduced by [95], Taylor-transformed skeletons allow for a more granular depiction of motion, enabling better recognition of actions that exhibit similar joint configurations but differ in motion dynamics, such as sitting *vs.*squatting. The Taylor-transformation method consists of computing the zeroth-order derivative (the original joint positions), the first-order derivative (velocity), and the second-order derivative (acceleration) for each joint in the skeleton sequence. These components capture the evolution of joint movements over time and offer richer information for action recognition models. However, while the transformation provides a more detailed temporal representation, it is often integrated into traditional models like ST-GCN, which may not fully capitalize on the additional information provided by these derivatives.

Our evaluation directly compares the performance of traditional skeleton sequences *vs.*Taylor-transformed skeleton sequences in the context of ST-GCN and Hyperformer. This side-by-side analysis allows us to better understand the impact of motion dynamics on action recognition, offering a clearer picture of when Taylor-transformed skeletons truly provide a benefit over static skeleton sequences. Furthermore, we analyze how different models handle these enriched representations, providing key insights into the advantages of motion-centric features.

**Hypergraph-based models.** Hypergraphs, as an extension of traditional graphs, allow for more complex relationships between entities. In the context of skeleton-based action recognition, hypergraph-based models have emerged as a way to better capture multi-joint interactions that cannot be modeled effectively with pairwise connections [29, 34, 37, 41, 57, 119]. The Hyperformer model [120] integrates hypergraphs with Transformer networks, enabling the model to learn high-order joint dependencies and coordinated movements more effectively. The introduction of hyperedges, which connect multiple joints in a single edge, helps capture the complex interactions that occur in actions such as dancing, sports, or complex gestures. The Hyperformer model operates by embedding skeleton sequences into a hypergraph structure and

then applying a Transformer-based self-attention mechanism to learn the temporal dependencies between the hyperedges. This structure enables the model to better capture joint dependencies that are not simply spatial (between two joints) but involve coordinated groupings of multiple joints, which is particularly useful for understanding human movements in dynamic contexts.

Our work evaluates Hyperformer alongside traditional graph-based models like ST-GCN, making a direct comparison between skeletal graphs *vs*.hypergraphs. We focus on evaluating how each model captures joint dependencies and motion dynamics. By examining the efficacy of hypergraphs in representing more intricate relationships between joints, we provide insights into the advantages and limitations of hypergraph-based models, particularly when applied to complex, coordinated actions. Additionally, our work contrasts skeleton *vs*.Taylor-transformed skeleton representations within these models, highlighting how enhanced motion representations influence performance in hypergraph-based models like Hyperformer.

**Hybrid approaches and data fusion.** Several approaches have explored combining graph-based models with other data modalities to improve skeleton-based action recognition [26, 49, 61, 74, 112]. For instance, methods that fuse visual features[19, 20, 25, 31, 54, 85, 90, 94, 99] (*e.g.*, from RGB cameras) with skeleton data have been proposed to capture both spatial and appearance-related features [18, 77, 86, 91, 115, 123]. These hybrid approaches rely on multi-stream learning frameworks or multi-modal fusion techniques, aiming to combine the advantages of skeleton data with rich appearance information [12, 53, 55, 58, 69, 70, 89, 108, 124]. However, while these methods may improve performance, they often introduce challenges in terms of data synchronization, computational complexity, and the alignment of features from different modalities [59, 72, 100].

Our focus remains on purely skeleton-based models, evaluating different representations and models without introducing additional complexity from other data sources. This allows us to isolate and assess the core factors influencing performance in skeleton-based action recognition, providing a clear comparison of skeleton *vs*.Taylor-transformed skeletons, graph *vs*.hypergraph representations, and the models that handle them.

**Insights and challenges.** A number of studies have investigated the challenges faced by skeleton-based models, such as handling occlusions, noise in joint estimations, and the difficulty of recognizing actions that are similar in appearance but differ in temporal dynamics [21, 47, 52, 65, 73, 76, 79, 83, 87, 88, 92, 93, 96]. While these works focus on improving robustness against these challenges, they tend to ignore the deeper structural choices that influence model performance. The choice of skeleton representation and the method of encoding motion dynamics are crucial in overcoming these challenges and improving model accuracy. Our paper directly addresses the importance of skeleton representations and the modeling of temporal dynamics, comparing traditional skeletons *vs*.Taylor-transformed skeletons and evaluating graph-based *vs*.hypergraph-based models. By doing so, we offer valuable insights into the inherent strengths and weaknesses of different representations and modeling strategies, providing a clearer path forward for future work in skeleton-based action recognition.

## 3 Key Aspects of Human Motion Modeling

Below, we discuss the key aspects of modeling human motion that are central to this paper (Figure 2). For our evaluation, we select the basic ST-GCN and Hyperformer models. These models serve as foundational frameworks upon which many existing approaches are built, offering simplicity and ease of experimentation while providing valuable insights into skeleton-based action recognition.

### 3.1 Skeletal Graph and Hypergraph

**Skeletal graphs and graph convolutions.** Skeletal graphs represent the human body as a network of nodes (joints) and edges (bones). This approach adheres to the body's natural anatomical structure. Each node represents a joint, and edges connect joints based on their physical links or spatial connectivity. Beyond spatial relationships, temporal edges are added to connect the same joint across different time steps, capturing the motion dynamics over time[38, 66, 110]. This combination of spatial and temporal edges creates a comprehensive structure for representing actions[1, 27, 28, 118]. GCNs, such as the widely used ST-GCN [110], process these skeletal graphs by aggregating information from neighboring nodes. The structure of the graph dictates how information flows during convolution, enabling the model to capture the motion of individual joints and their interactions with immediate neighbors. However, skeletal graphs inherently focus on pairwise relationships, which may miss complex interactions involving multiple joints simultaneously[119, 120].

**Skeletal hypergraphs and hypergraph convolutions.** Skeletal hypergraphs build upon the limitations of standard graphs by introducing hyper-edges, which connect groups of nodes rather than just pairs. This structure allows for modeling more complex relationships, such as the coordinated movement of multiple joints during an action. For example, in a *running* action, a hyper-edge could connect the hip, knee, and ankle to represent the interdependence of these joints in driving the motion. Hypergraph Convolutional Networks (HGCNs), like the Hyperformer, use these hyper-edges to aggregate features not just from neighboring nodes but from entire groups of related joints. This approach enables richer representations of actions, especially those requiring intricate coordination, such as gymnastics or team sports. By focusing on group-level interactions, hypergraphs can reveal subtle patterns that are missed in traditional graph-based models[109].

**Comparative insights.** The primary distinction between skeletal graphs and hypergraphs lies in their representation of relationships. Skeletal graphs are effective at modeling direct, pairwise connections and are well-suited for actions with clearly defined joint dependencies, such as walking or waving. However, they may oversimplify more complex actions where interactions among multiple joints play a critical role. In contrast, hypergraphs excel in capturing higher-order relationships by focusing on joint groups rather than pairs. This makes them particularly valuable for actions involving coordinated movements, such as *dancing* or *playing an instrument*. However, the increased complexity of hypergraphs requires careful design and computational resources, as determining which joints to group and how to weight their connections significantly impacts performance.

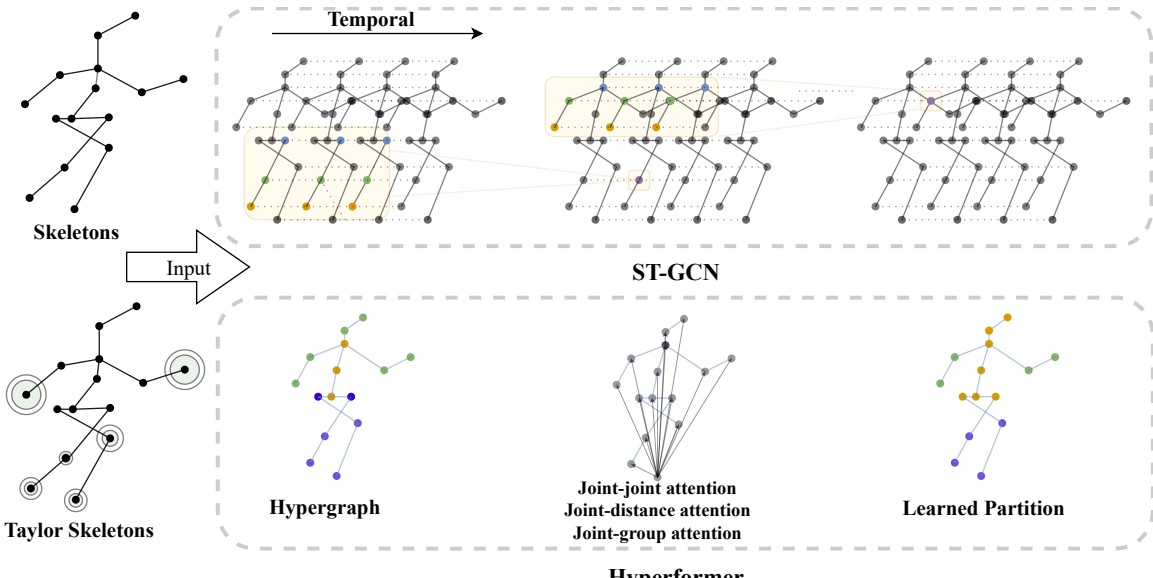

**Figure 2: The evaluation pipeline explores the performance of two state-of-the-art models, ST-GCN and Hyperformer. Both models are tested using original skeleton sequences, which emphasize spatial relationships, and Taylor-transformed skeletons, which highlight motion dynamics. This dual approach enables a comprehensive analysis of how spatial and temporal features impact action recognition performance, revealing distinct strengths and limitations for each model and data representation.**

## 3.2 Skeleton and Taylor-Transformed Skeleton

**Taylor-transformed skeletons** [95] offer a significant advancement by integrating motion dynamics directly into skeletal sequences. Inspired by the mathematical framework of Taylor series expansion, this method incorporates multiple temporal derivatives, such as velocity and acceleration, into the skeleton's representation. By adding higher-order derivatives, Taylor-transformed skeletons provide a more comprehensive and dynamic view of human actions, capturing not only the spatial configuration of the joints but also how they change over time. This approach highlights motion-centric features, which are crucial for differentiating between actions that may appear similar in terms of static body posture but differ in movement dynamics.

The transformation process begins with the standard skeletal sequence, where each frame includes the positions of body joints. The zeroth-order component represents the static positions of the joints. The first-order derivative, or velocity, is computed by subtracting consecutive joint positions, reflecting the rate of change in the joints' positions over time. The second-order derivative, or acceleration, captures the changes in velocity, providing deeper insights into the dynamics of movement transitions. Additional higher-order derivatives can be introduced to capture even more complex motion details. These temporal components are combined with the original skeleton, creating a representation that emphasizes dynamic motion rather than static configuration.

What makes Taylor-transformed skeletons stand out is their ability to prioritize temporal features that are essential for recognizing dynamic actions. Traditional skeletal representations [95] often focus on the spatial arrangement of joints, but they tend to overlook the temporal evolution of these relationships. By embedding motion-related features, Taylor-transformed skeletons ensure that action recognition models focus on meaningful temporal patterns, such as the acceleration of joints during specific movements. This approach reduces the reliance on redundant or less informative static data, allowing models to capture more nuanced motion characteristics.

**Comparative insights.** When integrated into graph-based models, such as ST-GCNs, Taylor-transformed skeletons can enhance model performance. The inclusion of motion-sensitive features improves the propagation of information through the graph, enabling more effective capture of temporal dependencies. In HGCNs, the transformed skeletons allow for a more refined modeling of complex relationships between joints, as hyper-edges can now represent dynamic groupings of joints that move together[35, 46]. This integration across different neural architectures underscores the versatility of Taylor-transformed skeletons in enhancing both spatial and temporal representations of human actions.

Despite their promising potential, Taylor-transformed skeletons also present challenges for future research. One challenge is the development of neural architectures that can fully use the higher-order temporal derivatives, especially when modeling more intricate and subtle motions. Additionally, adaptive methods for selecting the optimal level of Taylor expansion based on the complexity of the action could improve the efficiency and effectiveness of the transformation [95]. By addressing these challenges, Taylor-transformed skeletons could redefine motion representation in action recognition, offering a more accurate, interpretable, and dynamic approach to modeling human actions.

## 3.3 ST-GCN and Hyperformer

In the realm of skeleton-based action recognition, two prominent models have emerged: the ST-GCN [110] and the Hypergraph Transformer [120], known as Hyperformer. Both models aim to effectively capture the complex spatial and temporal dynamics inherent in human skeletal movements, yet they approach this challenge through distinct methodologies.

**ST-GCN.** Introduced in 2018, ST-GCN represents human skeletons as graphs, with joints as nodes and bones as edges. This graph-based representation allows the model to capture spatial dependencies between joints. To incorporate temporal dynamics, ST-GCN extends this graph structure across time, forming a spatiotemporal graph where each node connects to its temporal counterparts in adjacent frames. Graph convolutions are then applied to extract features that encapsulate both spatial configurations and their temporal evolutions. This design enables ST-GCN to model the intricate patterns of human motion effectively.

**Hyperformer.** Building upon the limitations of traditional graph-based models, Hyperformer introduces a hypergraph-based approach to capture higher-order relationships among joints. In a hypergraph, a single hyperedge can connect multiple nodes simultaneously, allowing the model to represent complex joint interactions that go beyond simple pairwise connections. Hyperformer integrates this hypergraph structure with transformer architectures, utilizing a novel Hypergraph Self-Attention mechanism. This mechanism enables the model to adaptively learn the importance of various joint groupings, effectively capturing both spatial and temporal dependencies without relying on a fixed topology.

**Comparative insights.** While both models aim to enhance action recognition by modeling skeletal data, they differ fundamentally in their representation and processing of joint relationships. ST-GCN relies on predefined graph structures based on human anatomy, which may limit its ability to adapt to the diverse and dynamic nature of human actions. In contrast, Hyperformer's use of hypergraphs allows for a more flexible representation, capturing complex joint interactions through higher-order connections. Additionally, the transformer-based architecture of Hyperformer facilitates adaptive learning of joint dependencies, potentially offering greater generalization across varied actions.

## 4 Experiment

### 4.1 Setup

**Datasets.** We choose the large-scale NTU-RGB+D 60 [71] and NTU-RGB+D 120 datasets [56] for the evaluations. The NTU-RGB+D 60 dataset, captured in a controlled laboratory environment with Kinect sensors, comprises around 56,000 video clips across 60 action classes. Each sample provides 3D joint coordinates for 25 body joints. Evaluations are performed using two standard benchmarks: the Cross-Subject benchmark, with 39,889 samples for training and 16,390 for testing, and the Cross-View benchmark, with 37,462 samples for training and 18,817 for testing.

NTU-RGB+D 120 extends the original dataset to include around 114,000 video clips spanning 120 action classes. This dataset introduces additional evaluation benchmarks: the Cross-Subject benchmark, with 63,026 samples for training and 25,883 for testing, and the Cross-Setup benchmark, which provides 54,471 training samples and 24,911 testing samples. The expanded dataset and benchmarks enable a more comprehensive evaluation of model performance.

**Metrics.** Recognition accuracies across all action classes are often visualized using a confusion matrix, which provides a detailed view of the algorithm's classification performance. The overall effectiveness of the algorithm on a given dataset is assessed by calculating the average recognition accuracy across all action classes. In our evaluation, we primarily report the top-1 recognition accuracy, representing the percentage of correctly classified samples where the top predicted label matches the ground truth. When required for comparative analysis, we also report the top-5 recognition accuracy, which accounts for cases where the correct label appears among the top five predictions.

**Models.** We use two models for evaluation: the ST-GCN and the Hyperformer. Both models are tested on the original skeleton sequences as well as their Taylor-transformed counterparts. For Taylor-transformed skeletons, we use the displacement concept with a single term, using four frames per temporal block and a step size of one across all datasets. No hyperparameter search, or skeleton sequence denoising is performed in this process.

The ST-GCN model begins with batch normalization, followed by nine spatial-temporal graph convolution layers. The network outputs 64 channels in the first three layers, 128 channels in the next three, and 256 channels in the final three, all using a temporal kernel size of 9. Residual connections are incorporated to improve stability and address gradient vanishing issues. Temporal strides of 2 are applied at the fourth and seventh layers to enable down-sampling, and dropout with a probability of 0.5 is applied after each unit to mitigate overfitting. After the final layer, global pooling generates a 256-dimensional feature vector, which is fed into a SoftMax classifier for action recognition. The model is trained using stochastic gradient descent with an initial learning rate of 0.01, which decays by 0.1 every 10 epochs.

The Hyperformer model is trained for 140 epochs using a cross-entropy loss function. The initial learning rate is set to 0.025 and decays by 0.1 at the 110th and 120th epochs. Training is conducted with a batch size of 64, and all sequences are resized to 64 frames. The model comprises a 10-layer architecture with 216 hidden channels, ensuring consistency across datasets and training conditions. These settings, along with the Taylor-transformed sequences, allow for a robust evaluation of action recognition performance. We use a batch size of 32 for ST-GCN and 128 for Hyperformer.

### 4.2 Evaluation

**Taylor skeletons are not always the best.** Table 1 presents the experimental results on NTU-60 and NTU-120. The results reveal that, overall, the Hyperformer model outperforms ST-GCN, both with and without the use of Taylor-transformed skeletons. This superior performance is attributed to the Hyperformer's ability to capture complex joint interactions through higher-order connections, whereas ST-GCN relies on predefined graph structures based on human physical connectivity, limiting its flexibility.

For the ST-GCN model, the use of Taylor-transformed skeletons significantly improves recognition accuracy on both NTU-60 and NTU-120. This demonstrates that Taylor skeletons enhance performance by introducing motion-sensitive features that better

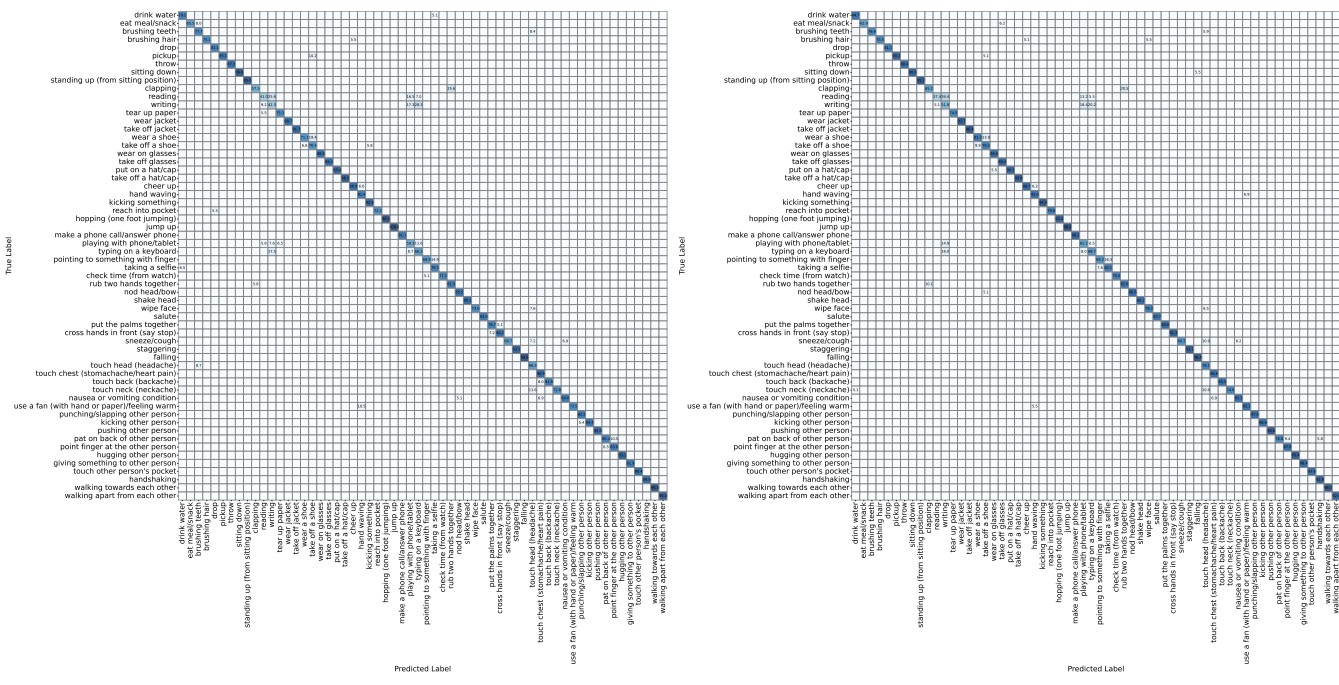

(a) Original skeletons.

(b) Taylor-transformed skeletons.

**Figure 3: Confusion matrices of ST-GCN on NTU-60 (X-Sub) using (a) original skeletons and (b) Taylor-transformed skeletons. Along the diagonal, darker colors represent higher recognition accuracy for each action class. Predictions below 5% are filtered out for clarity, with the complete confusion matrices available in the appendix. Taylor-transformed skeletons improve recognition accuracy for actions such as using a fan (with hand or paper), wearing a shoe, and touching head (headache). However, a decline in performance is observed for actions like hopping (one-foot jumping), sitting down, and touching chest (stomachache/heart pain). This drop may result from noise introduced by certain motion features, which negatively impact the model's performance. For a detailed examination, zooming in is recommended.**

**Table 1: Experimental results on NTU-60 and NTU-120 datasets, using both original skeleton sequences and Taylor-transformed skeletons evaluated with the ST-GCN and Hyperformer models.**

| | NTU-60 | | | | NTU-120 | | | |
| --- | --- | --- | --- | --- | --- | --- | --- | --- |
| | X-Sub | | X-View | | X-Sub | | X-Set | |
| | Top-1 | Top-5 | Top-1 | Top-5 | Top-1 | Top-5 | Top-1 | Top-5 |
| ST-GCN (w/o Taylor skeletons) | 81.5 | – | 88.3 | – | 70.7 | – | 73.2 | – |
| ST-GCN (w/ Taylor skeletons) | 83.5 | 97.2 | 89.4 | 98.7 | 74.1 | 93.0 | 75.8 | 93.4 |
| Hyperformer (w/o Taylor skeletons) | 90.7 | – | 95.1 | – | 86.6 | – | 88.0 | – |
| Hyperformer (w/ Taylor skeletons) | 86.7 | 97.4 | 92.1 | 99.0 | 79.1 | 95.0 | 81.9 | 95.7 |

propagate information through the graph structure, enabling more effective capture of temporal dependencies.

In contrast, for the Hyperformer model, using Taylor-transformed skeletons slightly decreases performance compared to the original skeletons. However, the Hyperformer still surpasses the ST-GCN model in all scenarios, both with and without Taylor skeletons. The reduced performance with Taylor skeletons may be due to the Hyperformer not being specifically designed to leverage this data format. While Taylor skeletons excel in representing motion dynamics, they lack detailed spatial information about joint arrangements and their interactions, which the Hyperformer may still rely on for optimal performance. These findings suggest that

while Taylor-transformed skeletons offer valuable motion-related features, they require new model architectures tailored to handle this data format effectively. Such advancements could further elevate the performance of skeletal action recognition tasks by combining the strengths of both motion and spatial information.

**Taylor skeletons highlight distinct motion dynamics in ST-GCN.** The confusion matrices for ST-GCN on NTU-60 (X-Sub) show distinct performance patterns between the use of original skeletons and Taylor-transformed skeletons (see Figure 3). Both approaches exhibit strengths and weaknesses, highlighting the trade-offs in using motion-sensitive transformations versus relying on spatially

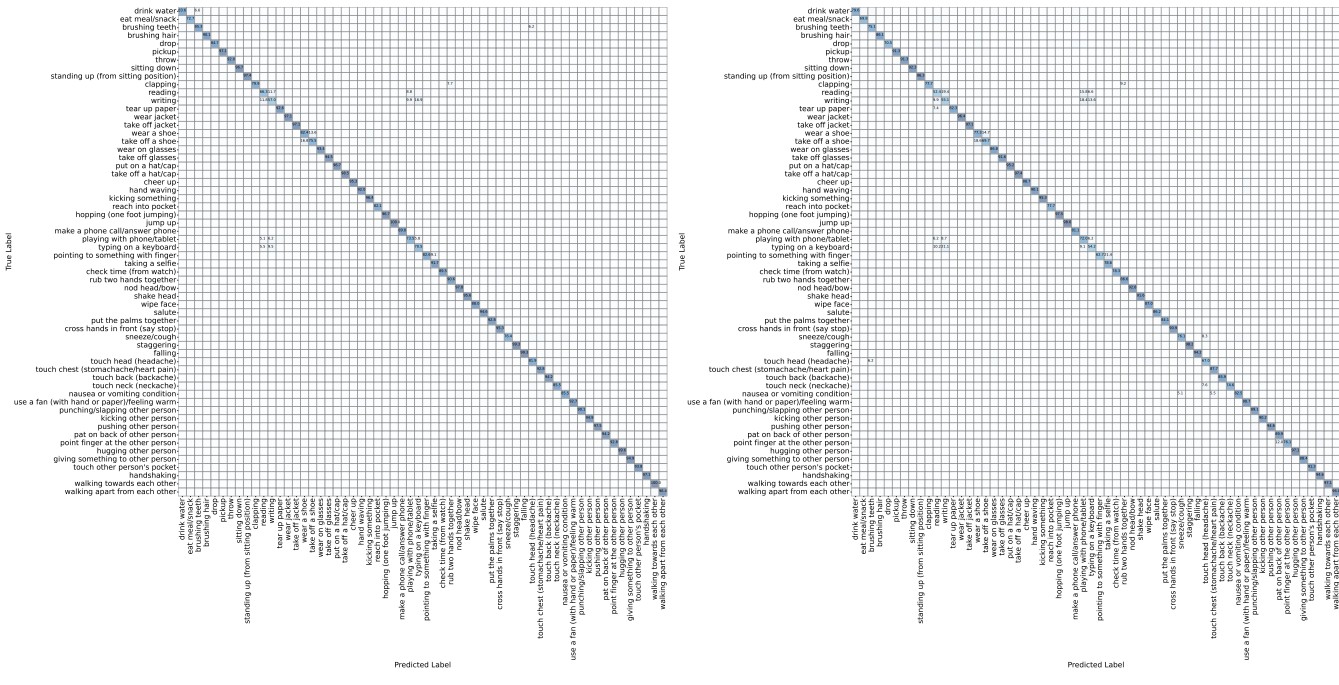

(a) Original skeletons.

(b) Taylor-transformed skeletons.

**Figure 4: Confusion matrices of Hyperformer on NTU-60 (X-Sub) using (a) original skeletons and (b) Taylor-transformed skeletons. Along the diagonal, darker colors represent higher recognition accuracy for each action class. Predictions below 5% are filtered out for clarity, with the complete confusion matrices available in the appendix. Taylor-transformed skeletons improve recognition accuracy for actions such as hopping and taking off jacket. However, a decline in performance is observed for actions like pointing to something with finger. This drop may result from noise introduced by certain motion features, which negatively impact the model's performance. For a detailed examination, zooming in is recommended.**

structured data. The Taylor-transformed skeletons enhance recognition accuracy for several dynamic actions, such as using a fan (with hand or paper)/feeling warm, wearing a shoe, and touching head (headache). These actions benefit from the motion-sensitive features introduced by the Taylor transformation, which emphasize temporal dynamics and fine-grained motion patterns. By capturing nuanced joint displacements, the Taylor skeletons improve the propagation of information through the graph structure, allowing the model to better understand subtle temporal dependencies.

Conversely, Taylor-transformed skeletons lead to noticeable performance declines for actions such as hopping (one-foot jumping) and sitting down. These actions often involve complex spatial interactions or are characterized by repetitive, cyclic motions. The Taylor transformation, while excellent at highlighting motion, de-emphasizes the spatial relationships and global structure of the skeleton. This shift can introduce noise, particularly in actions where spatial context plays a critical role in recognition.

The original skeleton sequences, despite lacking explicit motion emphasis, retain the full spatial arrangement of joints. This proves advantageous for actions heavily reliant on spatial cues, *e.g.*, walking apart from each other, where the relative positioning of joints is crucial. The ST-GCN model, with its predefined graph structures based on human physical connectivity, is well-suited to use such spatial information, enabling better performance on these tasks.

The comparative results suggest that while Taylor-transformed skeletons offer valuable insights into motion dynamics, they may not universally enhance recognition across all action types. Dynamic actions with distinct joint displacements are well-served by this transformation. However, actions requiring an understanding of spatial arrangements or involving subtle, repetitive motions may suffer from the loss of spatial information. This analysis also underscores the need for tailored model architectures capable of synergistically combining the strengths of motion-sensitive transformations and spatially structured data. Future research could explore hybrid models that integrate Taylor-transformed skeletons with spatially-aware features, potentially unlocking improved performance across diverse action categories.

**Which actions do Taylor skeletons improve performance on with the Hyperformer model?** The comparison between the confusion matrices of the Hyperformer model on NTU-60 (X-Sub) using original skeleton sequences and Taylor-transformed skeletons shows critical insights into the model's performance dynamics (see Figure 4). Each approach highlights distinct advantages and limitations, providing a nuanced understanding of how these data representations impact recognition across various action classes.

The Taylor-transformed skeletons enhance recognition accuracy for actions with prominent motion dynamics, such as hopping (one foot jumping). This improvement can be attributed to the Taylor transformation's ability to emphasize fine-grained motion patterns

and temporal changes, which are crucial for recognizing actions that rely heavily on dynamic motion cues. The Hyperformer, equipped with its ability to capture higher-order connections, benefits from this enriched temporal information, enabling a more detailed understanding of joint movements.

Despite these improvements, the Taylor-transformed skeletons lead to performance drops in certain actions, such as pointing to something with finger, which depend significantly on spatial configurations and global context. The Taylor transformation prioritizes motion over spatial arrangement, potentially diminishing the representation of joint relationships that are pivotal for recognizing these actions. This shift may introduce noise or obscure key spatial cues, leading to reduced accuracy in these categories.

The original skeleton sequences maintain a balanced representation of spatial and temporal features, which proves advantageous for actions requiring detailed spatial context. For instance, actions like pointing finger at the other person, which involve complex spatial interactions between body parts, are better captured using the original skeletons. The Hyperformer's architecture, while adaptable, appears to rely partially on this spatial information, which the original skeletons provide more effectively than the Taylor-transformed data. The results underscore the importance of tailoring model architectures to use specific data transformations effectively. The Taylor-transformed skeletons excel in representing dynamic actions but fall short in preserving spatial relationships, suggesting a gap in current model capabilities. Future advancements could involve hybrid architectures that integrate motion-sensitive features from Taylor transformations with spatially structured features from original skeletons. Such designs could achieve superior performance across a broader range of action classes. While the Taylor-transformed skeletons provide a unique perspective on motion dynamics, their effectiveness is action-dependent. The Hyperformer's performance indicates that optimizing the interplay between spatial and temporal features remains a critical direction for advancing action recognition tasks. A model capable of dynamically adjusting its reliance on spatial or temporal features based on the action type could significantly enhance recognition accuracy and robustness.

**ST-GCN *vs.*Hyperformer.** The comparison between ST-GCN and Hyperformer models, using both original skeleton sequences and Taylor-transformed skeletons on NTU-60 (X-Sub), reveals key distinctions in how these architectures handle spatial and temporal features. With original skeleton sequences, Hyperformer consistently outperforms ST-GCN across most action classes. This superiority stems from Hyperformer's advanced architecture, which effectively captures higher-order joint interactions and complex dependencies. Actions requiring detailed spatial reasoning, such as handshaking, wearing a jacket, and hugging, benefit significantly from Hyperformer's ability to integrate spatial and temporal features. In contrast, ST-GCN, reliant on predefined graph structures based on human physical connectivity, struggles with such nuanced interactions. However, ST-GCN still achieves reasonable accuracy, especially for actions characterized by clear spatial patterns.

When using Taylor-transformed skeletons, Hyperformer continues to outperform ST-GCN overall but exhibits a unique sensitivity to motion-focused transformations. The Taylor transformation enhances ST-GCN's recognition performance by introducing motion-sensitive features that improve information propagation through its graph structure. This improvement is evident in actions like drinking water, making a phone call, and reading, where dynamic motion cues are critical. Hyperformer, while maintaining higher accuracy than ST-GCN, shows performance drops for some actions with Taylor-transformed skeletons. Actions such as clapping, walking apart, and handshaking, which rely heavily on spatial context, are better recognized with original skeletons. The reduction in spatial detail from Taylor transformations diminishes Hyperformer's ability to fully use its higher-order connections, suggesting that its design still partially depends on spatially structured input.

The results underscore a trade-off between the two models. ST-GCN benefits from Taylor-transformed skeletons, which enhance its ability to capture temporal dependencies, narrowing the gap with Hyperformer for certain dynamic actions. However, Hyperformer remains superior in leveraging both spatial and temporal information, making it more robust across a wider range of action classes. This analysis highlights the need for hybrid approaches that can integrate the strengths of both architectures. Combining the motion sensitivity of Taylor transformations with the spatially aware design of Hyperformer could unlock further advancements in action recognition tasks, enabling models to adapt dynamically to the specific requirements of each action.

Additional experimental results and corresponding confusion matrices are provided in the Appendix.

## 5 Conclusion

We explore the effectiveness of traditional skeletons and Taylor-transformed skeletons for action recognition using ST-GCN and Hyperformer models on NTU-60 and NTU-120. Traditional skeleton sequences effectively capture spatial and temporal relationships but face limitations in distinguishing actions with subtle temporal variations. Taylor-transformed skeletons, by embedding motion dynamics, improve recognition for motion-intensive actions, though they introduce challenges for actions requiring detailed spatial context. In terms of model architectures, ST-GCN demonstrates robust performance with graph-based representations of human physical connectivity but struggles to capture higher-order joint interactions. Conversely, the Hyperformer uses hypergraph structures to model complex joint dependencies, outperforming ST-GCN overall, yet showing sensitivity to the loss of spatial information in Taylor-transformed skeletons. Key insights from our study include: (1) the potential of Taylor-transformed skeletons to enhance motion-sensitive recognition, (2) the advantages of hypergraph-based models like Hyperformer in representing higher-order dependencies, and (3) the critical interplay between motion dynamics and spatial structure in achieving optimal performance.

## Acknowledgments

Jushang Qiu conducted this research under the supervision of Lei Wang as part of his final year master's research project at ANU. This work was supported by the National Computational Merit Allocation Scheme 2024 (NCMAS 2024), with computational resources provided by NCI Australia, an NCRIS-enabled capability supported by the Australian Government.

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

## A    Additional Visualizations of Taylor Skeletons

Figure 5 presents an additional visual comparison between the original skeletons and the Taylor-transformed skeletons.

## B    Supplementary Results on NTU-60

Table 2, 3, 4, and 5 present additional experimental results on NTU-60.

**Table 2: Top 10 action classes showing the greatest accuracy gains and losses when comparing the performance of the ST-GCN model using original skeleton sequences versus Taylor-transformed skeleton sequences on NTU-60 (X-Sub).**

| Classes with Increased Accuracy | Original | Taylor skeletons | Difference |
|---|---|---|---|
| use a fan (with hand or paper)/feeling warm | 73.5 | 85.1 | ↑11.6 |
| wear a shoe | 71.1 | 81.3 | ↑10.2 |
| touch head (headache) | 66.3 | 76.1 | ↑9.8 |
| writing | 42.3 | 51.8 | ↑9.5 |
| make a phone call/answer phone | 81.1 | 90.2 | ↑9.1 |
| put the palms together | 79.7 | 88.8 | ↑9.1 |
| take off a shoe | 70.4 | 79.2 | ↑8.8 |
| clapping | 57.5 | 65.2 | ↑7.7 |
| reach into pocket | 72.3 | 79.6 | ↑7.3 |
| giving something to other person | 81.9 | 88.4 | ↑6.5 |
| **Classes with Decreased Accuracy** | **Original** | **Normalized Taylor** | **Difference** |
| hopping (one foot jumping) | 98.5 | 92.4 | ↓6.1 |
| sitting down | 96.0 | 90.8 | ↓5.2 |
| touch chest (stomachache/heart pain) | 90.9 | 86.6 | ↓4.3 |
| pat on back of other person | 80.4 | 76.8 | ↓3.6 |
| reading | 41.0 | 37.4 | ↓3.6 |
| walking apart from each other | 96.0 | 92.4 | ↓3.6 |
| put on a hat/cap | 93.4 | 90.1 | ↓3.3 |
| falling | 99.6 | 96.4 | ↓3.2 |
| shake head | 89.1 | 86.2 | ↓2.9 |
| eat meal/snack | 65.5 | 62.9 | ↓2.6 |

**Table 3: Top 10 action classes showing the greatest accuracy gains and losses when comparing the performance of the ST-GCN model using original skeleton sequences versus Taylor-transformed skeleton sequences on NTU-60 (X-View).**

| Classes with Increased Accuracy | Original | Taylor skeletons | Difference |
|---|---|---|---|
| typing on a keyboard | 62.7 | 72.2 | ↑9.5 |
| touch neck (neckache) | 85.1 | 90.8 | ↑5.7 |
| reading | 58.7 | 64.4 | ↑5.7 |
| cross hands in front (say stop) | 90.1 | 94.2 | ↑4.1 |
| pointing to something with finger | 87.6 | 91.1 | ↑3.5 |
| wear on glasses | 91.1 | 94.3 | ↑3.2 |
| brushing hair | 84.2 | 87.3 | ↑3.1 |
| use a fan (with hand or paper)/feeling warm | 90.8 | 93.7 | ↑2.9 |
| wipe face | 85.8 | 87.7 | ↑1.9 |
| make a phone call/answer phone | 87.0 | 88.9 | ↑1.9 |
| **Classes with Decreased Accuracy** | **Original** | **Taylor skeletons** | **Difference** |
| playing with phone/tablet | 83.2 | 72.2 | ↓11.0 |
| put the palms together | 93.4 | 83.9 | ↓9.5 |
| writing | 56.2 | 47.0 | ↓9.2 |
| take off a shoe | 87.3 | 79.1 | ↓8.2 |
| pat on back of other person | 85.4 | 78.5 | ↓6.9 |
| wear a shoe | 86.3 | 81.0 | ↓5.3 |
| taking a selfie | 90.8 | 85.8 | ↓5.0 |
| staggering | 97.8 | 93.0 | ↓4.8 |
| touch other person's pocket | 94.3 | 89.9 | ↓4.4 |
| drop | 95.9 | 92.1 | ↓3.8 |

**Table 4: Top 10 action classes showing the greatest accuracy gains and losses when comparing the performance of the Hyperformer model using original skeleton sequences versus Taylor-transformed skeleton sequences on NTU-60 (X-Sub).**

| Classes with Increased Accuracy | Original | Taylor skeletons | Difference |
|---|---|---|---|
| hopping (one foot jumping) | 96.7 | 97.5 | ↑0.8 |
| take off jacket | 97.1 | 97.1 | ↑0.0 |
| **Classes with Decreased Accuracy** | **Original** | **Taylor skeletons** | **Difference** |
| pointing to something with finger | 82.6 | 62.7 | ↓19.9 |
| point finger at the other person | 92.8 | 76.1 | ↓16.7 |
| typing on a keyboard | 70.5 | 54.2 | ↓16.3 |
| touch head (headache) | 81.9 | 67.0 | ↓14.9 |
| drop | 84.7 | 70.5 | ↓14.2 |
| reading | 66.3 | 52.4 | ↓13.9 |
| taking a selfie | 91.7 | 78.6 | ↓13.1 |
| check time (from watch) | 89.5 | 78.3 | ↓11.2 |
| touch neck (neckache) | 85.5 | 74.6 | ↓10.9 |
| tear up paper | 92.6 | 82.3 | ↓10.3 |

**Table 5: Top 10 action classes showing the greatest accuracy gains and losses when comparing the performance of the Hyperformer model using original skeleton sequences versus Taylor-transformed skeleton sequences on NTU-60 (X-View).**

| Classes with Increased Accuracy | Original | Taylor skeletons | Difference |
|---|---|---|---|
| take off a hat/cap | 98.4 | 98.7 | ↑0.3 |
| standing up (from sitting position) | 98.7 | 98.7 | ↑0.0 |
| **Classes with Decreased Accuracy** | **Original** | **Taylor skeletons** | **Difference** |
| writing | 74.6 | 51.1 | ↓23.5 |
| reading | 75.9 | 59.4 | ↓16.5 |
| touch head (headache) | 90.5 | 77.8 | ↓12.7 |
| typing on a keyboard | 77.8 | 65.2 | ↓12.6 |
| check time (from watch) | 96.2 | 85.4 | ↓10.8 |
| drop | 96.2 | 87.3 | ↓8.9 |
| brushing hair | 97.2 | 88.6 | ↓8.6 |
| brushing teeth | 94.3 | 85.8 | ↓8.5 |
| tear up paper | 95.9 | 87.7 | ↓8.2 |
| eat meal/snack | 92.4 | 84.8 | ↓7.6 |

## C    Supplementary Results on NTU-120

Table 6, 7, 8, and 9 present additional experimental results on NTU-120.

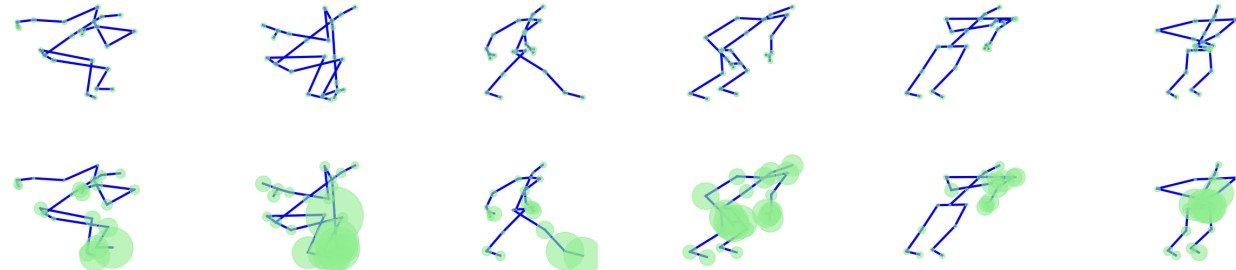

**Figure 5: Visual comparison of (*top*) original skeletons and (*bottom*) Taylor-transformed skeletons. From left to right, the depicted actions are: "take off a shoe", "wear a shoe", "kicking something", "nausea or vomiting", "sneeze/cough", and "touch chest (stomachache/heart pain)". Taylor-transformed skeletons are overlaid on the original skeletons, with green circles indicating the intensity of motion.**

**Table 6: Top 10 action classes showing the greatest accuracy gains and losses when comparing the performance of the ST-GCN model using original skeleton sequences versus Taylor-transformed skeleton sequences on NTU-120 (X-Sub).**

| Classes with Increased Accuracy | Original | Taylor skeletons | Difference |
|---|---|---|---|
| playing with phone/tablet | 49.8 | 57.5 | ↑7.7 |
| tennis bat swing | 67.9 | 75.1 | ↑7.2 |
| exchange things with other person | 78.6 | 85.2 | ↑6.6 |
| hit other person with something | 47.1 | 53.2 | ↑6.1 |
| reading | 44.0 | 49.8 | ↑5.8 |
| take something out of a bag | 70.8 | 76.4 | ↑5.6 |
| cross hands in front (say stop) | 81.2 | 86.6 | ↑5.4 |
| side kick | 83.3 | 87.8 | ↑4.5 |
| move heavy objects | 74.5 | 78.9 | ↑4.4 |
| wear jacket | 91.6 | 95.3 | ↑3.7 |
| **Classes with Decreased Accuracy** | **Original** | **Taylor skeletons** | **Difference** |
| play magic cube | 54.7 | 31.6 | ↓23.1 |
| cutting paper (using scissors) | 49.7 | 27.6 | ↓22.1 |
| take a photo of other person | 84.7 | 68.2 | ↓16.5 |
| put the palms together | 85.3 | 73.9 | ↓15.6 |
| open bottle | 60.4 | 45.0 | ↓15.4 |
| brushing teeth | 87.2 | 74.7 | ↓12.5 |
| walking apart from each other | 96.7 | 85.5 | ↓11.2 |
| wield knife towards other person | 55.0 | 44.4 | ↓10.6 |
| follow other person | 94.4 | 84.2 | ↓10.2 |
| step on foot | 87.0 | 77.2 | ↓9.8 |

**Table 7: Top 10 action classes showing the greatest accuracy gains and losses when comparing the performance of the ST-GCN model using original skeleton sequences versus Taylor-transformed skeleton sequences on NTU-120 (X-Set).**

| Classes with Increased Accuracy | Original | Taylor skeletons | Difference |
|---|---|---|---|
| move heavy objects | 73.7 | 85.5 | ↑11.8 |
| kicking other person | 79.5 | 89.6 | ↑10.1 |
| make victory sign | 40.6 | 47.6 | ↑7.0 |
| writing | 31.1 | 37.8 | ↑6.7 |
| put on bag | 78.4 | 84.7 | ↑6.3 |
| take off headphone | 80.6 | 86.2 | ↑5.6 |
| hit other person with something | 51.9 | 57.2 | ↑5.3 |
| running on the spot | 88.8 | 94.1 | ↑5.3 |
| cross arms | 81.4 | 86.5 | ↑5.1 |
| high-five | 84.6 | 89.0 | ↑4.4 |
| **Classes with Decreased Accuracy** | **Original** | **Taylor skeletons** | **Difference** |
| pat on back of other person | 90.1 | 71.0 | ↓19.1 |
| thumb up | 64.3 | 48.2 | ↓16.1 |
| cutting paper (using scissors) | 41.0 | 25.4 | ↓15.6 |
| check time (from watch) | 81.9 | 66.8 | ↓15.1 |
| put the palms together | 85.3 | 70.8 | ↓14.5 |
| typing on a keyboard | 71.3 | 53.4 | ↓13.7 |
| walking apart from each other | 97.4 | 83.3 | ↓14.1 |
| playing with phone/tablet | 67.1 | 53.4 | ↓13.7 |
| brushing teeth | 86.4 | 75.8 | ↓10.6 |
| pointing to something with finger | 73.3 | 63.1 | ↓10.2 |

**Table 8: Top 10 action classes showing the greatest accuracy gains and losses when comparing the performance of the Hyperformer model using original skeleton sequences versus Taylor-transformed skeleton sequences on NTU-120 (X-Sub).**

| Classes with Increased Accuracy | Original | Taylor skeletons | Difference |
|---|---|---|---|
| eat meal/snack | 70.5 | 71.3 | ↑0.8 |
| punching/slapping other person | 85.4 | 85.8 | ↑0.4 |
| take off jacket | 98.6 | 98.6 | ↑0.0 |
| writing | 52.6 | 52.6 | ↑0.0 |
| **Classes with Decreased Accuracy** | **Original** | **Taylor skeletons** | **Difference** |
| cutting paper (using scissors) | 68.8 | 36.3 | ↓32.5 |
| move heavy objects | 95.1 | 63.6 | ↓31.5 |
| open a box | 77.7 | 55.1 | ↓22.6 |
| point finger at the other person | 92.0 | 72.1 | ↓19.9 |
| sniff (smell) | 84.2 | 65.2 | ↓19.6 |
| shake fist | 82.3 | 63.5 | ↓18.8 |
| cutting nails | 67.0 | 48.3 | ↓18.7 |
| open bottle | 76.8 | 58.5 | ↓18.3 |
| reading | 64.5 | 46.5 | ↓18.0 |

**Table 9: Top 10 action classes showing the greatest accuracy gains and losses when comparing the performance of the Hyperformer model using original skeleton sequences versus Taylor-transformed skeleton sequences on NTU-120 (X-Set).**

| Classes with Increased Accuracy | Original | Taylor skeletons | Difference |
|---|---|---|---|
| touch other person's pocket | 86.7 | 89.3 | ↑2.6 |
| handshaking | 94.8 | 95.6 | ↑0.8 |
| throw | 89.0 | 89.6 | ↑0.6 |
| **Classes with Decreased Accuracy** | **Original** | **Taylor skeletons** | **Difference** |
| cutting paper (using scissors) | 63.5 | 34.8 | ↓28.7 |
| shoot at other person with a gun | 82.7 | 56.9 | ↓25.8 |
| point finger at the other person | 94.4 | 69.4 | ↓25.0 |
| move heavy objects | 94.0 | 69.9 | ↓24.1 |
| play magic cube | 71.6 | 50.3 | ↓21.3 |
| make victory sign | 62.2 | 41.8 | ↓20.4 |
| staple book | 52.7 | 33.8 | ↓18.9 |
| wield knife towards other person | 73.2 | 55.1 | ↓18.1 |
| open a box | 75.1 | 58.0 | ↓17.1 |
| check time (from watch) | 89.7 | 73.2 | ↓16.5 |

## D    Full Confusion Matrix Visualizations

Figure 6, 7, 8, 9, 10, 11, 12, and 13 display the complete confusion matrix visualizations for NTU-60 and NTU-120.

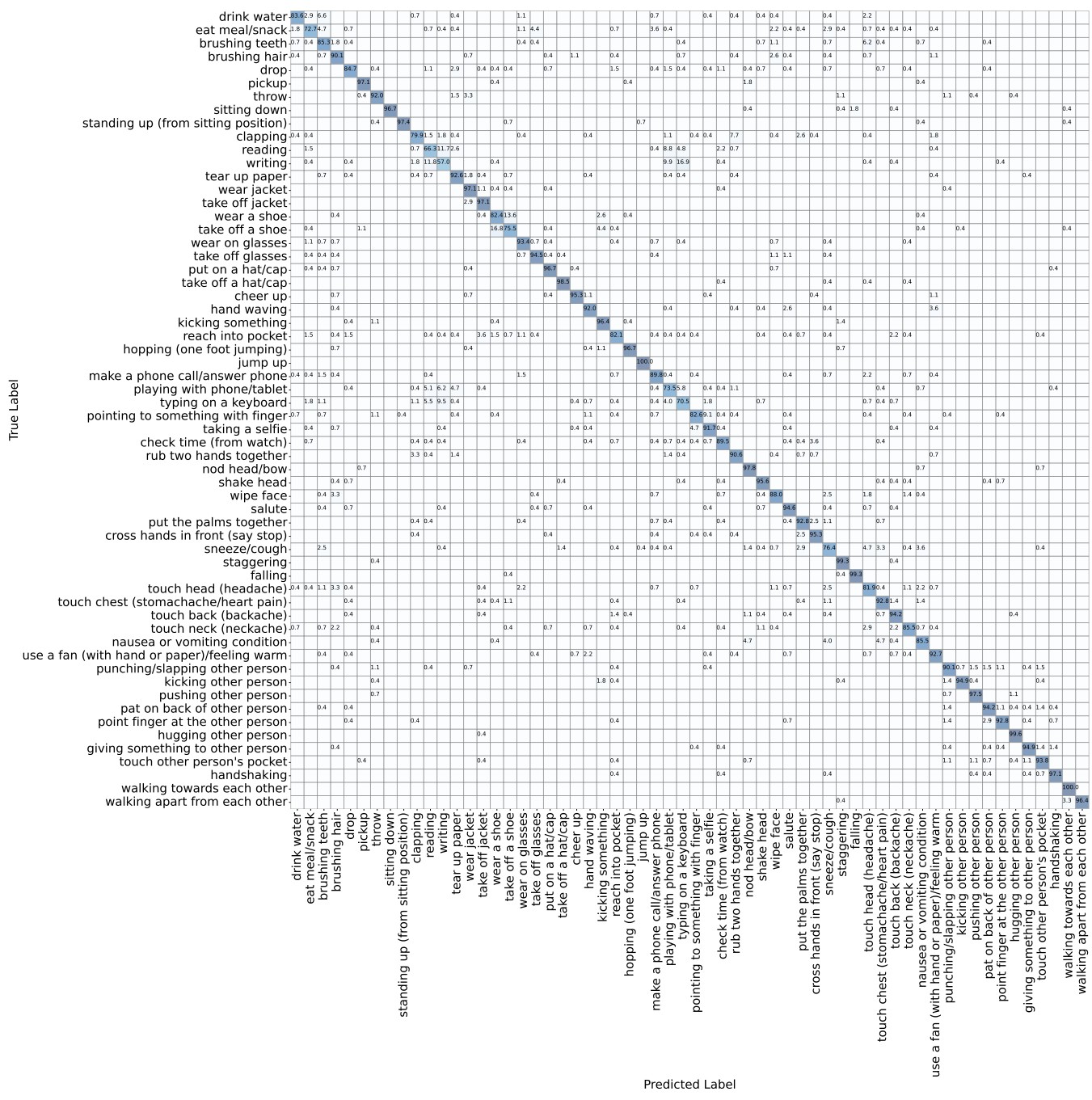

**Figure 6: Confusion matrix of ST-GCN on NTU-60 (X-Sub) using original skeletons. Along the diagonal, darker colors represent higher recognition accuracy for each action class. For a detailed examination, zooming in is recommended.**

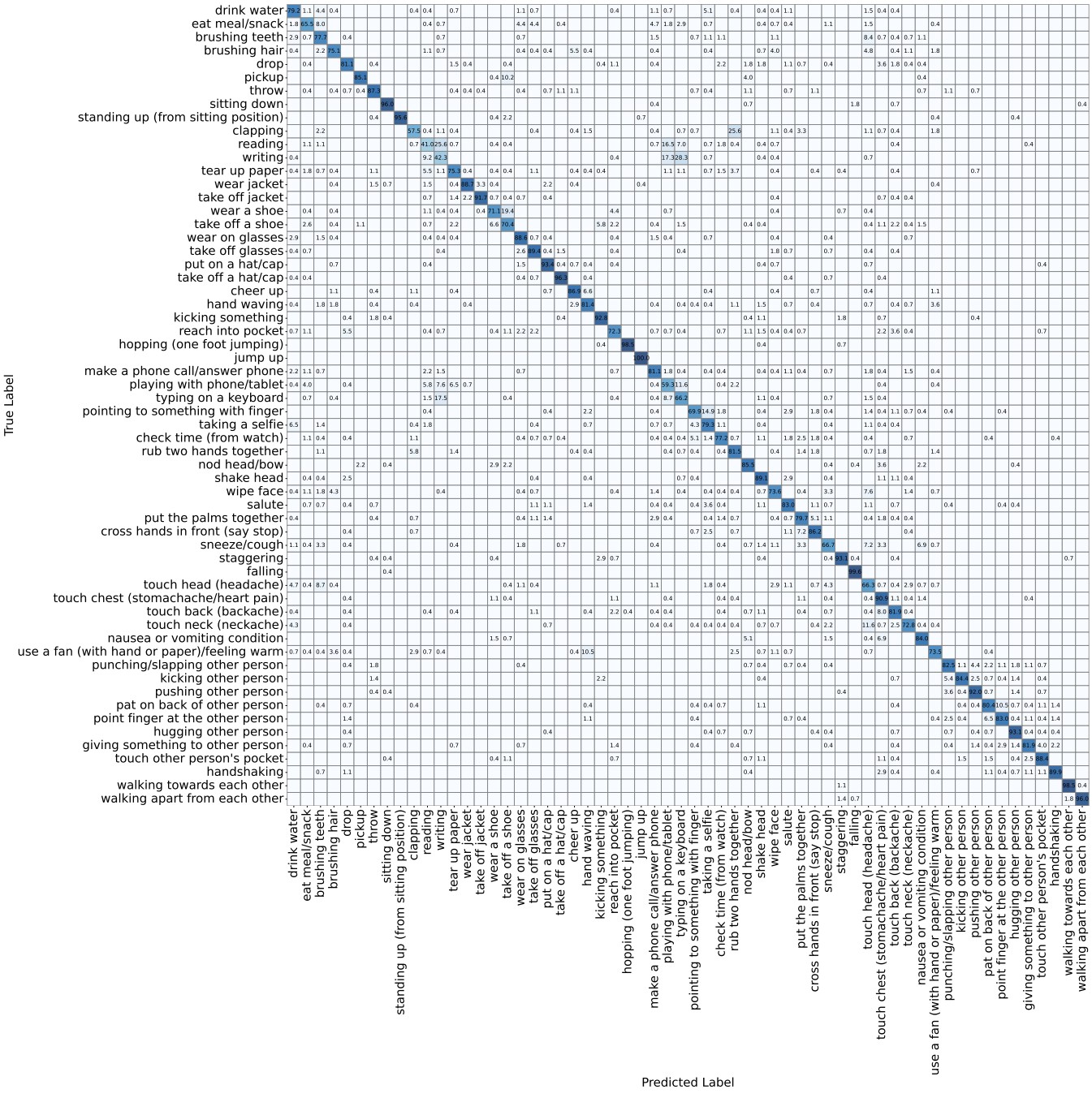

**Figure 7: Confusion matrix of ST-GCN on NTU-60 (X-Sub) using Taylor-transformed skeletons. Along the diagonal, darker colors represent higher recognition accuracy for each action class. For a detailed examination, zooming in is recommended.**

 

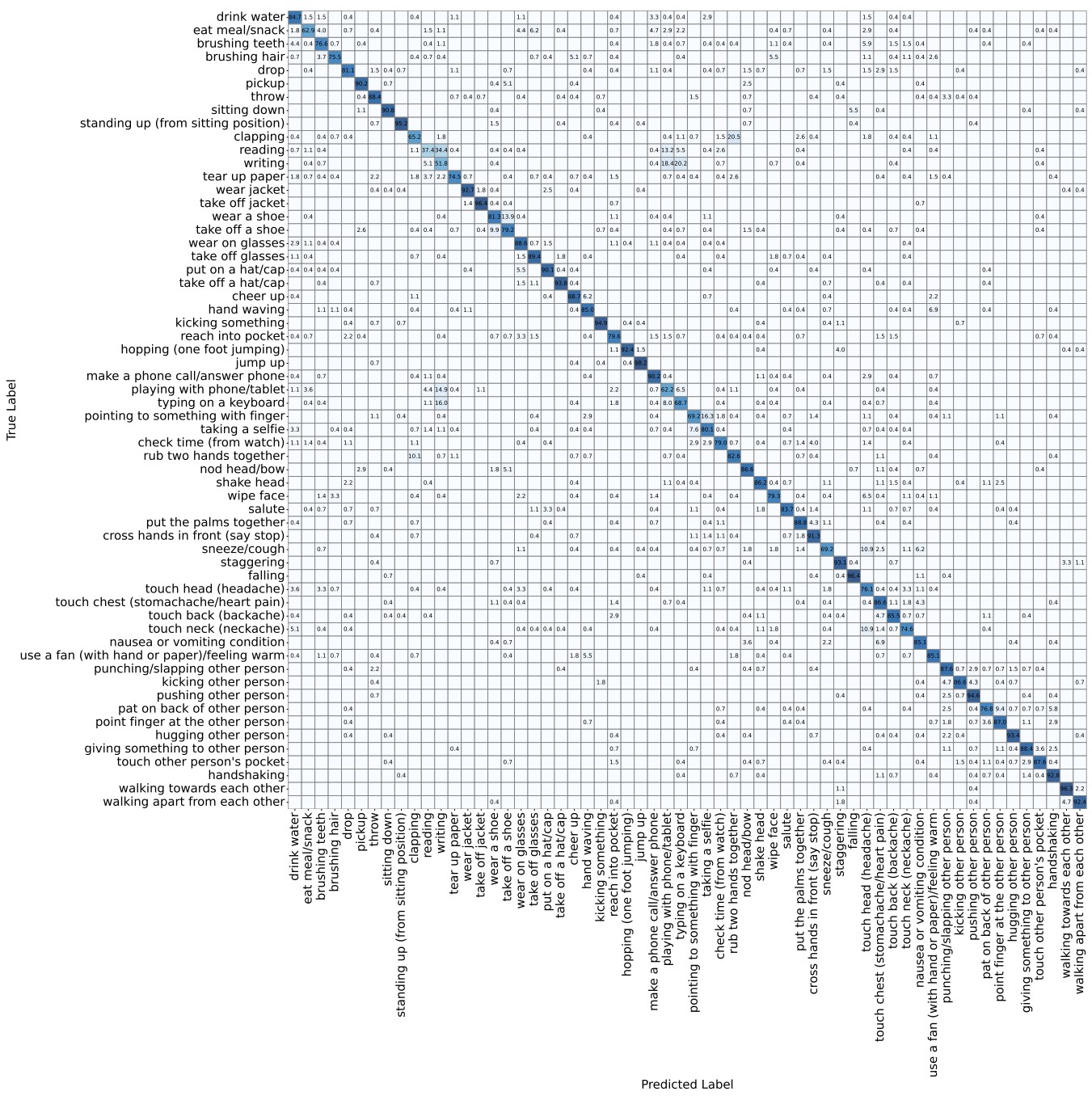

**Figure 8: Confusion matrix of Hyperformer on NTU-60 (X-Sub) using original skeletons. Along the diagonal, darker colors represent higher recognition accuracy for each action class. For a detailed examination, zooming in is recommended.**

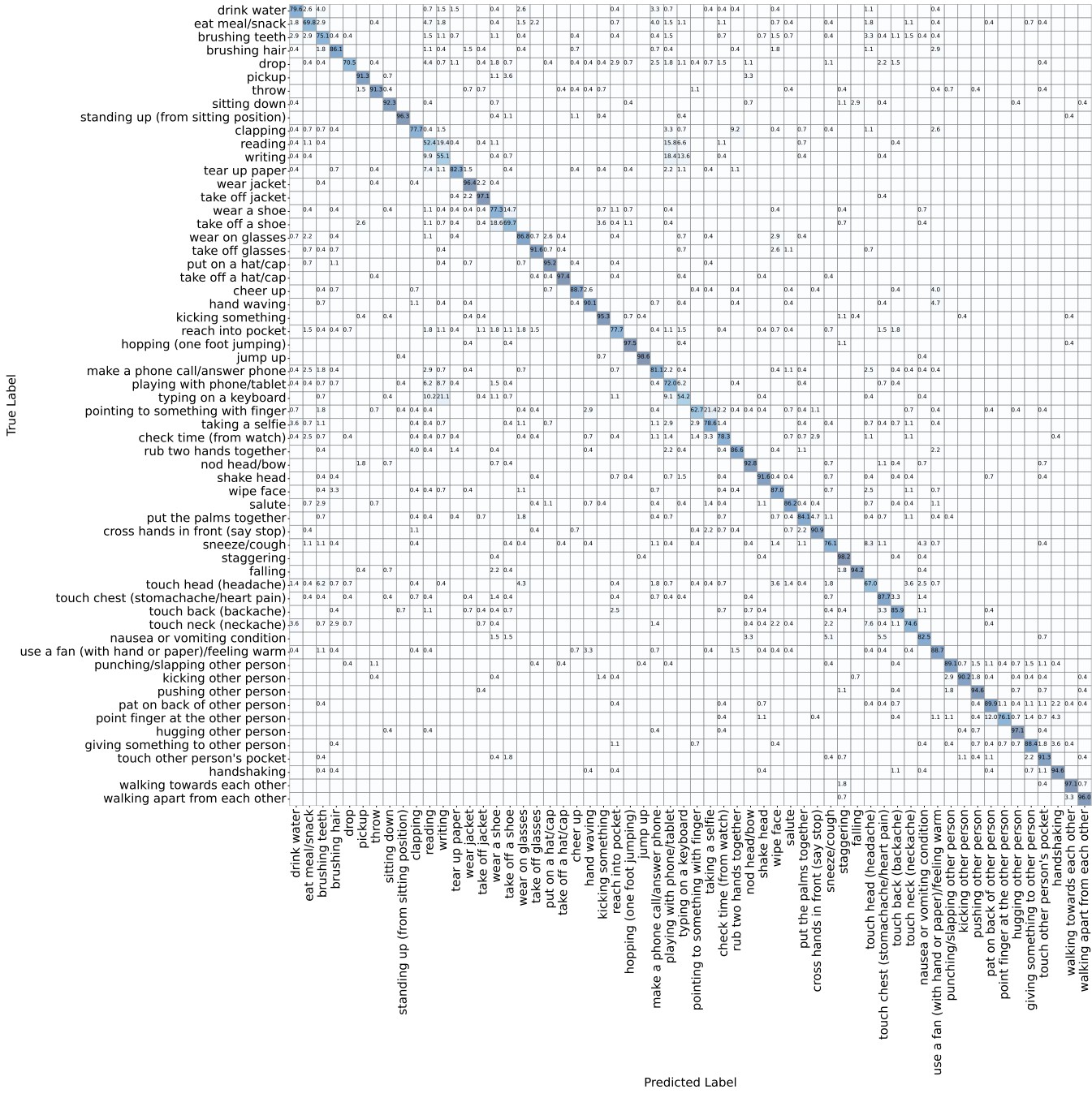

**Figure 9: Confusion matrix of Hyperformer on NTU-60 (X-Sub) using Taylor-transformed skeletons. Along the diagonal, darker colors represent higher recognition accuracy for each action class. For a detailed examination, zooming in is recommended.**

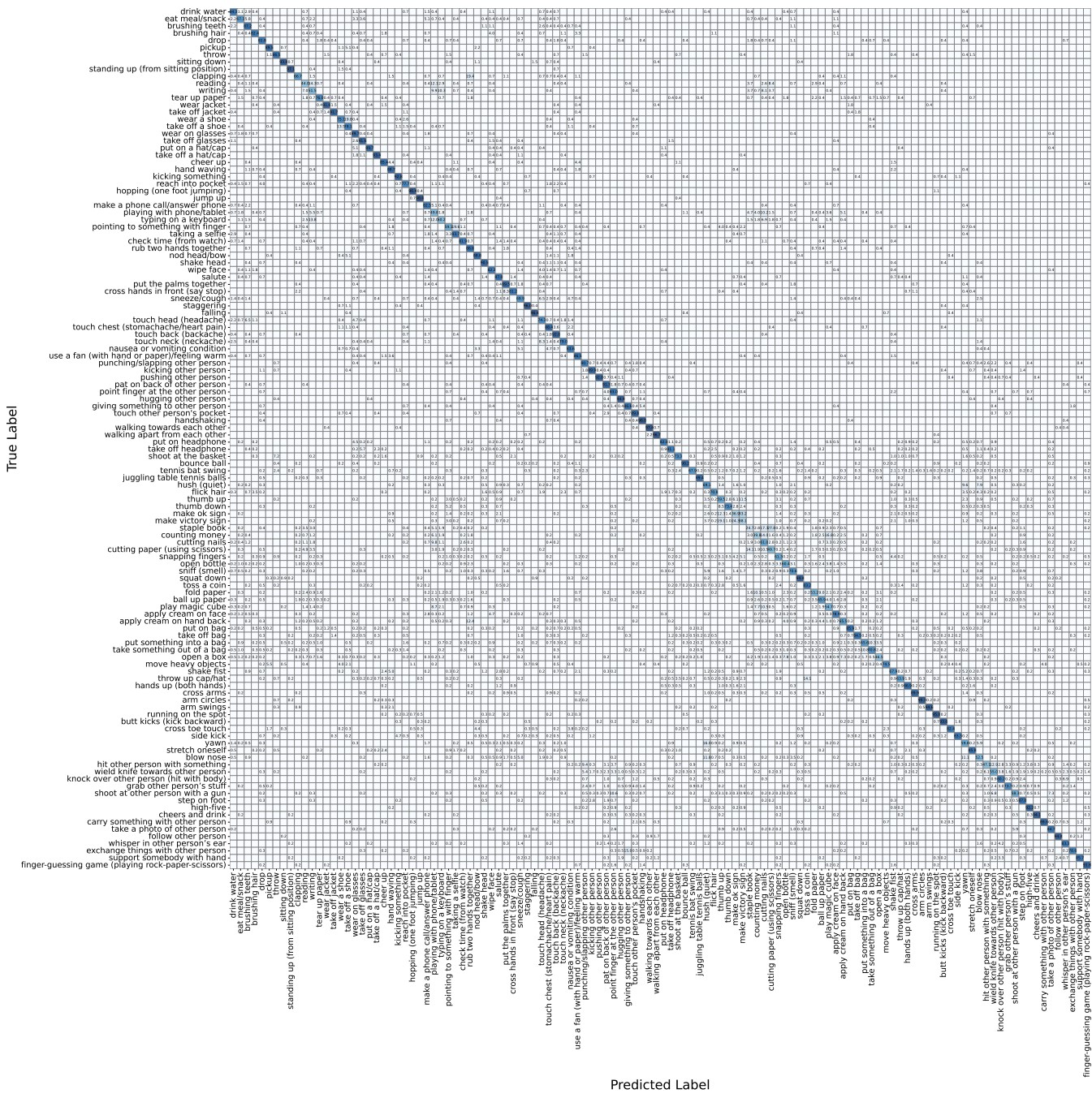

**Figure 10: Confusion matrix of ST-GCN on NTU-120 (X-Sub) using original skeletons. Along the diagonal, darker colors represent higher recognition accuracy for each action class. For a detailed examination, zooming in is recommended.**

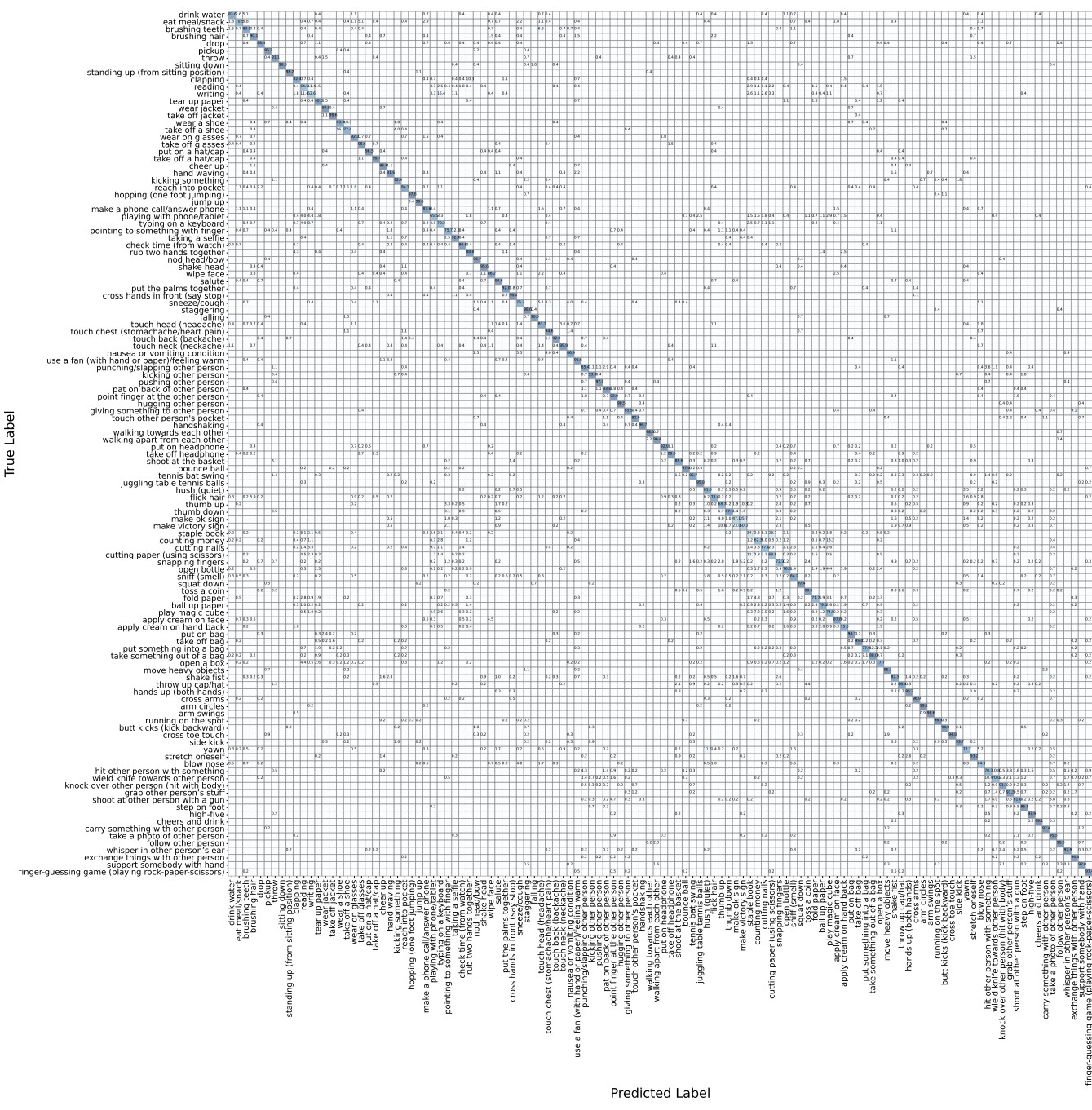

**Figure 11: Confusion matrix of Hyperformer on NTU-120 (X-Sub) using original skeletons. Along the diagonal, darker colors represent higher recognition accuracy for each action class. For a detailed examination, zooming in is recommended.**

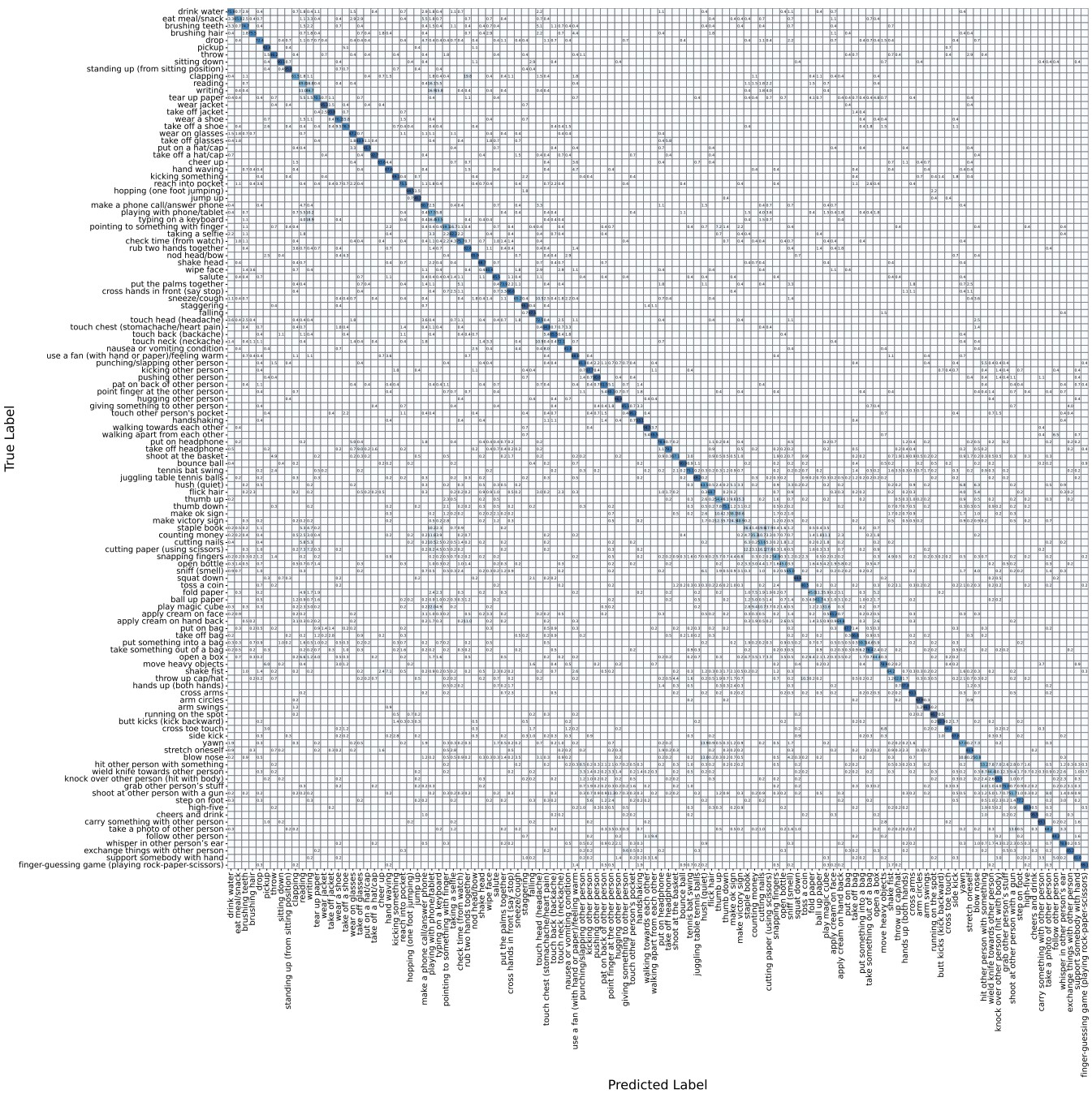

**Figure 12: Confusion matrix of ST-GCN on NTU-120 (X-Sub) using Taylor skeletons. Along the diagonal, darker colors represent higher recognition accuracy for each action class. For a detailed examination, zooming in is recommended.**

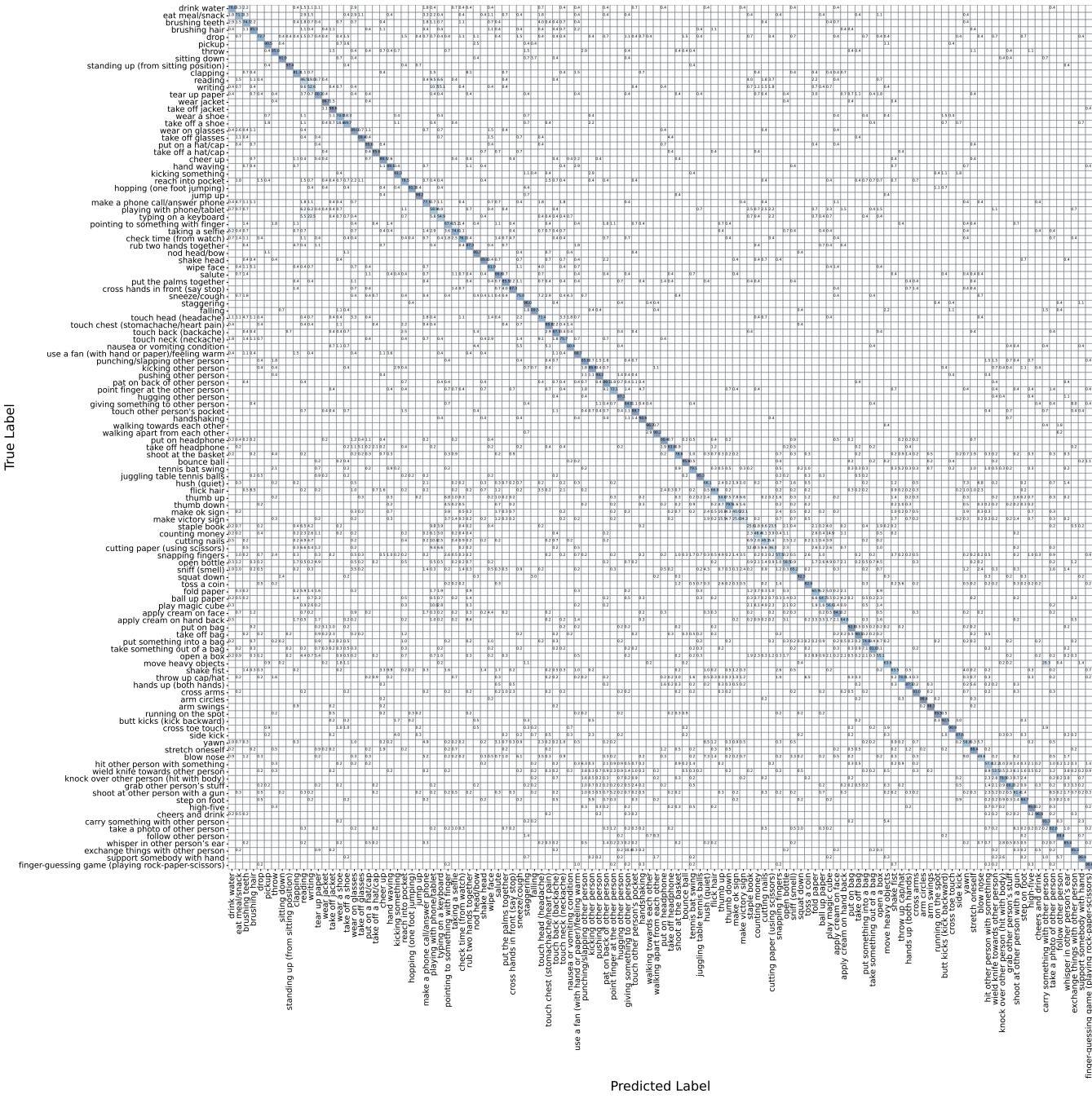

**Figure 13: Confusion matrix of Hyperformer on NTU-120 (X-Sub) using Taylor skeletons. Along the diagonal, darker colors represent higher recognition accuracy for each action class. For a detailed examination, zooming in is recommended.**