# OpenReview forum: "Evolving Skeletons: Motion Dynamics in Action Recognition"
_ACM.org/TheWebConf/2025/Workshop/TIME — TIME 2025 Poster_

### Official Review · Reviewer_Wnvj · 2025-01-11
**Minor Revisions**

**Rating:** 6
**Confidence:** 4

**Review:**

Strengths
1. Good abstract
2. Good in depth literature review with experimental setup
3. Detailed results

Weakness
1. Mathematical & emperical share out - equations and analysis

Recommendations - Accept with minor revisions

---

### Official Review · Reviewer_15mq · 2025-01-19
**The article provides a basic summary of skeleton-based action re-identification, focusing on graph methods and skeleton sequences, but it is not a comprehensive survey or technical paper.**

**Rating:** 4
**Confidence:** 3

**Review:**

After a meticulous review of the entire text, despite its rich content, this article primarily serves as a basic summary within the domain of skeleton-based action re-identification, focusing on graph methods and skeleton sequences. It falls short of being considered a comprehensive survey or a technical paper, for the following reasons:
- Limited Motivation Lacking Evidence. Mentioned in line 144, the motivation behind the study is quite limited and not well-supported by evidence. The experimental approach, which involves applying various graph methods to different skeleton sequences, appears more akin to a case study rather than a demonstration of technological innovation.
- Narrow Scope and Lack of Depth. In the third section, the article compares differences between skeleton graphs, hypergraphs, ST-GCN, and Taylor-Transformed Skeletons. However, the scope of this comparison is limited and fails to adequately classify or delve deeply into the body of work on skeleton-based action recognition, offering only superficial summaries and comparisons.
- Casual Layout and Formatting Issues. The article's layout appears casual with insufficient attention to detail, such as excessive white space at line 154 and inappropriate placement of Figure 2.

---

### Official Review · Reviewer_gJye · 2025-01-19
**Review of Evolving Skeletons: Motion Dynamics in Action Recognition**

**Rating:** 5
**Confidence:** 4

**Review:**

This paper comprehensively summarizes skeleton-based action recognition from the perspectives of graph neural networks and skeleton sequences. Specifically, it contrasts methods such as skeletal graphs, ST-GCN, and hyperformer to showcase the evolution of graph neural networks in this field. It also emphasizes the importance of dynamic features by comparing skeleton and Taylor-transformed skeleton, providing insightful observations for each method. Extensive experiments were conducted from these two perspectives to validate the approaches.

Main Strengths:
- The paper provides a detailed account of the development of skeletal graphs and presents unique insights.
- The author conducted plenty of experimental validations.

Main Weaknesses:
- The motivation of the study is weak in line 143. (1) there is insufficient evidence supporting this motivation; (2) the paper does not demonstrate significant differences in previous methods across various domains and contexts.
- The backbone frameworks selected for the study, such as ST-GNN and hyperformer, are not particularly novel and fail to convince.
- The writing and logical structure of the paper, especially in the Introduction, lacks clarity, making it difficult to discern the connection between the first three paragraphs and the motivation.
- The layout of the article needs improvement,  for example, excessive whitespace at line 154 and inappropriate placement of Figure 2.

---

### Official Review · Reviewer_rNDX · 2025-01-23

**Rating:** 7
**Confidence:** 3

**Review:**

Paper is critically analyzing the interplay of static and dynamic pose representations with different skeleton modeling techniques and providing Model-specific insights into the strengths and limitations of ST-GCN and Hyperformer for dynamic and static pose modeling.
Author choose the large-scale NTU-RGB+D 60 and NTURGB+D 120 datasets to further analysed  the methodology.
The core contributions primarily involve comparing existing models providing actionable research directions, emphasizing the development of hybrid models that can seamlessly integrate motion and spatial features for improved action recognition which is an incremental work.

---

### Meta-Review · Area_Chair_uPpJ · 2025-01-26

**Recommendation:** Accept (Oral)
**Confidence:** 3

**Metareview:**

The paper is very well put together and provides detailed explanations of the use of ST-GCN and the Hyperformer model for action recognition tasks. From the perspective of AI techniques, such as graph-based learning and multi-task optimization, the methodology is solid, leveraging large and well-known datasets like NTU-60 and NTU-120 for validation. However, the paper’s focus on domain-specific applications like motion recognition and action detection, while important, is outside my core expertise.

Overall, the implementation of graph-based learning models is clear with an innovative use of Taylor transformed motion derivates. While the application of AI is solid, the implementation focuses heavily on domain-specific considerations, which are not directly aligned with my professional expertise.

This paper’s focus on motion recognition and its use cases, particularly in areas like surveillance and sports analytics, is outside my area of expertise. My review primarily focuses on the AI techniques and data integration aspects, which are competently implemented and demonstrate technical rigor.

Based on these observations, while the work has merit, I would rank it as Accept (Poster) due to its technical rigor but limited generalizability beyond its domain-specific scope.

---

### Decision · Program_Chairs · 2025-01-27

**Decision:**

Accept (Poster)

**Comment:**

The program chair concurs with the area chair's decision.

For the camera-ready version, please revise your paper according to the feedback provided by the reviewers.

Workshop papers must be written in English, follow a double-column format, and comply with the [ACM template](https://www2025.thewebconf.org/short-papers) and formatting guidelines. The template is also available in [Overleaf](https://www.overleaf.com/latex/templates/association-for-computing-machinery-acm-sig-proceedings-template/bmvfhcdnxfty). For authors using Microsoft Word, the Word Interim Template is recommended.

Camera-ready versions of accepted papers can and should include all information to identify authors, and should acknowledge any funding received that directly supported the presented research.

In addition, ensure that the DOI (to be provided by the PCs at a later stage) is included, and cite the workshop (to appear) using the following reference:

```
@inproceedings{time2025,
  title={TIME 2025: 1st International Workshop on Transformative Insights in Multi-faceted Evaluation},
  author={Lei Wang and Md Zakir Hossain and Syed Islam and Tom Gedeon and Sharifa Alghowinem and Isabella Yu and Serena Bono and Xuanying Zhu and Gennie Nguyen and Nur Haldar and Seyed Jalali and Abdur Razzaque and Imran Razzak and Rafiqul Islam and Shahadat Uddin and Naeem Janjua and Aneesh Krishna and Manzur Ashraf},
  booktitle={ACM Web Conference Workshop},
  year={2025}
}
```

Please note that at least one in-person registration is required for each accepted workshop paper to be included in the Companion Proceedings of WWW 2025. All accepted papers must be presented at the conference. Papers not presented (no-shows) may be withdrawn from the companion proceedings. Presentations will be conducted in two formats: oral and poster.

The camera-ready deadline for workshop papers is 7 February 2025 (AoE).